

# Symbiosis revisited: phosphorus and acid buffering stimulate N₂ fixation but not *Sphagnum* growth

Eva van den Elzen[1*], Martine A.R. Kox[2], Sarah F. Harpenslager[1], Geert Hensgens[1], Christian Fritz[1], Mike S.M. Jetten[2], Katharina F. Ettwig[2], Leon P.M. Lamers[1]

5  [1]Department of Aquatic Ecology & Environmental Biology, Institute for Water and Wetland Research, Radboud University, Heyendaalseweg 135, 6525 AJ Nijmegen, the Netherlands
[2]Department of Microbiology, Institute for Water and Wetland Research, Radboud University, Heyendaalseweg 135, 6525 AJ Nijmegen, the Netherlands

*Correspondence:* Eva van den Elzen (e.vandenelzen@science.ru.nl)



**Abstract**

In pristine *Sphagnum* dominated peatlands, (di)nitrogen ($N_2$) fixing (diazotrophic) microbial communities associated with *Sphagnum* mosses contribute substantially to the total nitrogen input, increasing carbon sequestration. The rates of symbiotic nitrogen fixation reported for *Sphagnum* peatlands, are, however, highly variable and experimental work on regulating

factors that can mechanistically explain this variation is largely lacking. For two common fen species (*Sphagnum palustre* and *S. squarrosum*) from a high nitrogen deposition area (25 kg N $ha^{-1}$ $y^{-1}$), we found that diazotrophic activity (as measured by $^{15-15}N_2$ labeling) was still present. This was surprising, given that nitrogen fixation is a costly process. We tested the effects of phosphorus availability and buffering capacity by bicarbonate rich water, mimicking a field situation in fens with stronger groundwater or surface water influence, as potential regulators of nitrogen fixation rates and *Sphagnum*

performance. We expected that the addition of phosphorus, being a limiting nutrient, would stimulate both diazotrophic activity and *Sphagnum* growth. We indeed found that nitrogen fixation rates were doubled. Plant performance, in contrast, did not increase. Raised bicarbonate levels also enhanced nitrogen fixation, but had a strong negative impact on *Sphagnum* performance. These results explain the higher nitrogen fixation rates reported for minerotrophic and more nutrient-rich peatlands. The contrasting effects of phosphorus and bicarbonate on *Sphagnum* spp and their diazotrophic communities

reveal strong differences in optimal niche for both partners with respect to conditions and resources. This suggests a trade-off for the symbiosis of nitrogen fixing microorganisms with their *Sphagnum* hosts, in which a sheltered environment apparently outweighs the less favorable environmental conditions. We conclude that microbial activity is still nitrogen limited under eutrophic conditions because dissolved nitrogen is being monopolized by *Sphagnum*. Moreover, the fact that diazotrophic activity can significantly be upregulated by increased phosphorus addition and acid buffering, while *Sphagnum*

spp do not benefit, reveals remarkable differences in optimal conditions for both symbiotic partners and questions the concept of a direct mutualism.

Key-words: plant-microbiome interactions, ecophysiology, symbiosis, diazotrophy, peatland, fen, bicarbonate, pH, nitrogen deposition, nutrients

**1. Introduction**

Nitrogen (N) availability is considered to limit or co-limit primary production in pristine *Sphagnum*-dominated ecosystems (Aerts et al., 1992; Lamers et al., 2000; Limpens and Berendse, 2003). Peat mosses (*Sphagnum* spp.) function as a filter that very effectively absorbs particularly ammonium ($NH_4^+$) but also nitrate ($NO_3^-$) from atmospheric deposition, leading to N limitation in the rhizosphere of vascular plants (Lamers et al., 2000; Bragazza et al., 2004; Fritz et al., 2014). Since the

availability of N determines primary production, there appears to be a close link between the N and C cycles (Hungate et al., 2003; Vitousek et al., 2013). This link is especially important in peatlands, which, by storing substantial amounts of C, play an important role in global C cycling (Ruesch and Gibbs, 2008; Clymo and Hayward, 1982). Being ecosystem engineers in



peatlands, *Sphagnum* spp. produce recalcitrant litter, rich in phenolic compounds (Verhoeven and Toth, 1995), and actively acidify their environment (Clymo and Hayward, 1982). This, combined with moist, anaerobic conditions results in the accumulation of peat with a high C content (Van Breemen, 1995). Recently, it has been shown that the high $N_2$ fixation activity of the *Sphagnum* microbiome could explain the discrepancy between low inputs of atmospheric N and high N

accumulation rates in the peat of pristine *Sphagnum* peatlands (Vile et al., 2014), confirming the strong link between C and N accumulation. On the other end, high atmospheric N deposition may also compromise the C sequestration function of peatlands by stimulating microbial processes such as overall decomposition (Bragazza et al., 2006) and denitrification (Gruber and Galloway, 2008).

$N_2$ fixing microorganisms (diazotrophs) live on the surface and inside dead hyaline cells of *Sphagnum* (Opelt et al., 2007; Bragina et al., 2012; Larmola et al., 2014), forming a symbiosis with their host. A highly diverse microbial community, including Proteobacteria, Verrucomicrobia and Cyanobacteria has been found to colonize peat mosses (Bragina et al., 2014) and many of these microorganisms have the capacity to fix $N_2$ (Bragina et al., 2013; Kox et al., 2016). Also in other bryophytes, like *Hylocomiaceae* (feather mosses) such a symbiotic relationship can be found with $N_2$ fixing cyanobacteria,

supplying up to 50% of the total N input in boreal forests (Rousk et al., 2013). These phototrophic diazotrophs provide N to their host in exchange for C compounds (Bay et al., 2013; Leppänen et al., 2013). In these moss symbioses, as well as in vascular plant symbioses, application of high rates of inorganic N were found to decrease $N_2$ fixation rates, with the host plant shifting to the use of this readily available inorganic N source (Gundale et al., 2011; Zackrisson et al., 2004; Rousk et al., 2014). Although the exact nature of the *Sphagnum*-microorganism symbiosis remains unknown, N fixed by

cyanobacteria associated with *Sphagnum* was found to enhance *Sphagnum* growth (Berg et al., 2013). A high variation in rates of $N_2$ fixation has not only been found for different species and different systems, but also for similar ecosystem types at different locations. To our knowledge, the mechanistic explanation for this high variation of symbiotic $N_2$ fixation rates in *Sphagnum* peatlands is still lacking.

In areas with high N deposition like in our field sites, the necessity for microorganisms with diazotrophic capacity to actually fix $N_2$ can be expected to diminish, as $NH_4^+$ availability usually leads to down-regulation of the expression of the nitrogenase enzyme responsible for $N_2$ fixation (Dixon and Kahn, 2004). Other nutrients than N have been suggested to influence $N_2$ fixation, especially phosphorus (P) (Vitousek and Field, 1999) which is generally the second nutrient limiting primary production (Bieleski, 1973; Vance, 2001). P limitation has been shown to play an important role in biomass growth and

functioning of peatlands (Larmola et al., 2013; Hill et al., 2014; Fritz et al., 2012) and appeared to control $N_2$ fixation rates (Toberman et al., 2015; Vitousek et al., 2002; Chapin et al., 1991). Besides, isolated cyanobacteria were shown to be directly stimulated by P (Mulholland and Bernhardt, 2005). In peat mosses from N-rich sites, increased P availability can be expected to complement the high N supply (Limpens et al., 2004) and lead to an increase in moss growth. We therefore hypothesized that the addition of P would improve the performance of the *Sphagnum*-microorganism association in high N



deposition areas. In addition, in view of a mutualistic relationship between the moss and its diazotrophs, we expected that higher $N_2$ fixation rates might provide additional N that, combined with higher P availability, would increase *Sphagnum* photosynthesis and growth even further, as long as no other resource or condition becomes limiting.

Next to nutrient availability, the alkalinity and pH of the environment is known to be a key biogeochemical factor affecting *Sphagnum* presence and performance in peatlands (mires). Higher concentrations of bicarbonate ($HCO_3^-$) and concomitantly higher pH values (from 7.5 and upwards), through the influence of minerotrophic groundwater or surface water in rich fens, have been shown to hamper *Sphagnum* growth (Clymo, 1973; Lamers et al., 1999). While the effect of environmental factors such as pH and nutrient availability on *Sphagnum* itself has been thoroughly studied (Clymo, 1973; Kooijman and Paulissen,

2006; Bragazza and Gerdol, 2002), it remains unknown how these environmental factors influence the activity of its diazotrophic community and how this in turn affects *Sphagnum* performance in peatlands. Information about the factors regulating the diazotrophic community is vital to understand the high variation in $N_2$ fixation rates in *Sphagnum* dominated wetlands that may strongly affect both nutrient and carbon cycling.

We therefore used a controlled, full-factorial set-up to experimentally test the effects of P and $HCO_3$ addition on $N_2$ fixation rates of the diazotrophic community and on photosynthesis and growth of two common fen species, *Sphagnum squarrosum* Crome and *S. palustre* L. from a Dutch rich fen. Our prime research question was whether P availability and alkalinity were key regulators of both diazotrophic and *Sphagnum* activity. In this way, we were also able to explore the nature of the symbiotic interaction, i.e. which benefits or costs the diazotrophic microbial community would experience through the close

association with their host, and vice versa.

## 2. Methods

### 2.1 Collection of *Sphagnum* and peat

Two common species of *Sphagnum*, *S. squarrosum* and *S. palustre* were chosen for their widespread occurrence (Europe, America, Asia, Australia), and their differences in habitat preference. While both are typical fen species, *S. squarrosum* is

known to withstand slightly more buffered (higher pH) conditions (Clymo, 1973; Rydin and Jeglum, 2006). To mimic their natural habitat, including moist conditions and supply of substrate-derived $CO_2$ for *Sphagnum* development (Smolders et al., 2001), peat mosses were placed on *Sphagnum* peat monoliths. Both peat mosses and monoliths were collected from the peatland Ilperveld in the Netherlands (52°26'22.68"N; 4°56'54.81"E), where monoliths (25 x 12 x 20 cm depth) were placed in glass mesocosms (25 x 12 x 30 cm depth) and then transported to the lab. Soils were kept wet with demineralized water (1

cm above soil level) and allowed to acclimatize for 2 weeks. Patches of 70 (*S. palustre*) or 80 (*S. squarrosum*) capitula (top 2 cm of moss) representing similar fresh weights were placed on top of the monoliths. A total of 16 mesocosms were placed in a water bath maintained at 15° C (using a cryostat) with a light regime of 16h light using four 400 watt lamps (Hortilux





Schreder HS2000, Monster, the Netherlands) and one growth lamp with 120 deep red/white LEDs (Philips, GreenPower LED, Poland), providing in total 150 µmol PAR m$^{-2}$ s$^{-1}$ and a temperature of 18°C at vegetation level. The light level was chosen to mimic realistic field conditions where *Phragmites australis* and sedges in these fens create low, but not limiting light levels for *Sphagnum* spp (Bonnett et al., 2010; Kotowski and Diggelen, 2004).

**2.2 Experimental set up**

After acclimatization, there was a constant flow of different treatment solutions through the mesocosms, at a flow rate of 5.4 L per week using peristaltic pumps (Masterflex L/S tubing pump; Cole-Parmer, Schiedam, the Netherlands) to create constant conditions in a 1 cm water layer over the soils. The lower 1 cm of *Sphagnum* spp. was flooded, while capitula were just above the water layer. Four different treatment solutions were applied (N=4 replicates per treatment), which were

spatially distributed in a randomized block design. The treatments were applied in a full factorial design with a P treatment of 10 µmol L$^{-1}$ P (as Na$_4$P$_2$O$_7$) and a HCO$_3$ treatment of 3 mmol L$^{-1}$ NaHCO$_3$. Also 5 mg L$^{-1}$ of sea salt with small amounts of trace elements (Tropic Marine, aQua united LTD, Wartenberg, Germany) was added to all treatment solutions (including control) to mimic rainwater quality and prevent osmotic stress. Furthermore, each mesocosm was provided with an amount of rainwater equivalent to the mean annual rainfall in the Netherlands (750 mm) and with an N concentration equivalent to

the Dutch atmospheric deposition of 25 kg N ha$^{-1}$ y$^{-1}$. Three times a week, 150 ml of artificial rainwater was sprayed on the peat mosses, containing 5 mg L$^{-1}$ sea salt (Tropic Marine, aQua united LTD, Wartenberg, Germany), 19 µmol L$^{-1}$ KCl, 10 µmol L$^{-1}$ CaCl$_2$, 10 µmol L$^{-1}$ Fe-EDTA, 1 µmol L$^{-1}$ KH$_2$PO$_4$, 0.7 µmol L$^{-1}$ ZnSO$_4$, 0.8 µmol L$^{-1}$ MnCl$_2$, 0.2 µmol L$^{-1}$ CuSO$_4$, 0.8 µmol L$^{-1}$ H$_3$BO$_3$, 8 nmol L$^{-1}$ (NH$_4$)$_6$Mo$_7$O$_{24}$ and 91 µmol L$^{-1}$ NH$_4$NO$_3$.

**2.3 Plant performance**

Photosynthetic rates of the mosses were determined using a fast greenhouse gas analyzer with cavity ringdown spectroscopy (GGA-24EP; Los Gatos Research, USA). From each mesocosm one individual of each moss species was taken and placed in a closed glass vile (100 ml) at similar light conditions as used in the experimental set up, connected to the gas analyzer. Changes in CO$_2$ concentrations were measured over a time period of 5 minutes, in a closed loop with the gas analyzer. Additionally, dark measurements were carried out for each sample, and gross photosynthetic rates were calculated by

correcting the slope of CO$_2$ decrease in light with the slope of the CO$_2$ increase in dark. Also, capitula were counted and average lengths of *Sphagnum* individuals determined. Total fresh weight (FW) of *Sphagnum* biomass was measured, after which material was dried at 70°C for 48 hours to determine dry weight (DW) in order to calculate relative growth rates.

**2.4 N$_2$ fixation rates and elemental composition of *Sphagnum***

Two subsamples (the top 2 cm of two individuals) of *S. squarrosum* and *S. palustre* from each mesocosm were placed

separately in 30 ml glass serum bottles with rubber stoppers. 6 ml of headspace was removed with an injection needle and replaced with $^{15-15}$N$_2$ gas (98 atom% $^{15}$N, Sigma-Aldrich, Germany), leading to 20% $^{15}$N$_2$ labeling. Samples were incubated





for 48 hours with a light regime of 16 hours of light (150 μmol m$^2$ s$^{-1}$ PAR) at 18° C. They were then dried at 70° C for 48 hours and ground using a mixer mill (MM301, Retsch, Germany) for 2 minutes at 30 rotations s$^{-1}$. Total N concentrations and isotopic ratios were determined using an elemental analyzer (Type NA 1500 Carlo Erba, Thermo Fisher Scientific Inc., USA) coupled online via an interface (Finnigan Conflo III) to a mass-spectrometer (Thermo Finnigan DeltaPlus, USA). For

every control and P-treated sample an additional incubation was carried out under similar but dark conditions. For every incubated subsample a control sample was taken that had not been incubated with $^{15-15}$N$_2$, to correct for background isotopic composition as influenced by the different treatments. The corrected increases in $^{15}$N labeling were converted to N$_2$ fixation rates (nmol N$_2$ gDW$^{-1}$ h$^{-1}$), using the average of both labeled subsamples. These N$_2$ fixation rates were also converted to rates of N fixed per unit area with bulk density data from the field (dry weight of upper 2 cm of each species in a 10 cm$^2$ plot

(N=4 replicates)). Fixation rates per hectare per year were calculated assuming N$_2$ fixation activity throughout the growing season (Rousk et al., 2015) during a growing season of around 250 days for peatlands in the northern hemisphere with mild winters (Helfter et al., 2015; Zhu et al., 2012) and corrected for an average seasonal temperature of 13° C, assuming a Q10 of three (Kravchenko and Doroshenko, 2003; Granhall and Selander, 1973; Alexander and Schell, 1973).

Total P and potassium (K) concentrations were determined in digestates of dried and ground *Sphagnum*-microorganism tissue. Digestates were prepared by heating in 500 μl HNO$_3$ (65%) and 200 μl H$_2$O$_2$ (30%) for 16 min in a microwave (mls 1200 Mega, Milestone Inc., Sorisole, Italy). After dilution with demineralized water, P and K concentrations were measured by inductively-coupled plasma emission spectrometry (IRIS Intrepid II, Thermo Electron corporation, Franklin, MA, USA).

### 2.5 Soil and water chemistry

At the end of the experiment, two soil subsamples of a fixed volume were taken from each mesocosm. Homogenized subsamples were dried at 70° C for 72 hours and weighted to determine bulk densities. Organic matter concentrations were determined through loss on ignition at 550° C for 3 hours. Dried soils were digested with 4 ml HNO$_3$ (65%) and 1 ml H$_2$O$_2$ (30%) using a microwave and measured by inductively-coupled plasma emission spectrometry as described above. C and N contents of dried soil were measured using an elemental analyzer (see above). Soil properties can be found in Table 1.

The pH of surface water was measured with a standard Ag/AgCl electrode (Orion Research, Beverly, USA) combined with a pH meter (Tim840 titration manager; Radiometer analytical, Lyon, France). Alkalinity was determined by titrating down to pH 4.2 with 0.1 N HCl using an auto burette (ABU901 Radiometer, Copenhagen, Denmark). Concentrations of PO$_4^{3-}$, NO$_3^-$ and NH$_4^+$ were measured colorimetrically with a 3 Auto Analyzer system (Bran & Luebbe, Norderstedt, Germany), using

ammonium molybdate (Henriksen, 1965), hydrazine sulfate (Kamphake et al., 1967) or salicylate (Grasshoff and Johannsen, 1972), Cl was determined with a Technicon Flame Photometer IV Control (Bran & Luebbe, Norderstedt, Germany). Concentrations of Al, Ca, Fe, S, Mg, Mn, Na, P and K were analyzed by inductively coupled plasma spectrometry (see above).



### 2.6 Statistical analyses

Values displayed in bar graphs are means ± standard error (SEM) (N=4). To test for the effect of P, $HCO_3^-$ and different species on different parameters three-way ANOVA's were used, using P, $HCO_3^-$ and species as independent variables (fixed factors). Normality was tested with a Shapiro-Wilk test on the residuals of the ANOVA and data that were not normally

distributed were log-transformed prior to analysis to meet conditions of parametric tests. Homogeneity of the data was checked with Levene's test of equality of variances. No interaction effects were found for any of the parameters and significance was accepted at a confidence level of $P < 0.05$. Statistical tests were performed using IBM SPSS Statistics 21.0 (IBM Corporation, 2012).

### 3 Results

From our full factorial experiment with additions of P and/or $HCO_3$ we took measurements on surface water (water quality changes) and on *Sphagnum*-microorganism tissue: $N_2$ fixation activity, plant performance parameters and nutrient contents.

### 3.1 Water quality changes

The addition of P (10 µmol $L^{-1}$) resulted in an increase in total P in the surface water (F = 6.044; $P < 0.05$) from 0.7 µmol $L^{-1}$ to a concentration of 6.0 µmol $L^{-1}$, indicating net uptake and/or binding of P. Supply of $HCO_3^-$ increased pH (from 4.3 to 8.0)

and alkalinity (from 0.1 to 2.8 meq $L^{-1}$) in the surface water (F=2780.292; $P<0.001$). Furthermore, upon addition of $HCO_3^-$ the concentrations of $NH_4$, Ca, Mg, Cl, S, Fe and Al in the water increased two to five times, and K concentration was increased by a factor 1.4 (Table 2).

### 3.2 $N_2$ fixation

Under light conditions, diazotrophic activity was similar for both *Sphagnum* spp. Control incubations showed high average

$N_2$ fixation rates of around 40 nmol N $gDW^{-1}$ $h^{-1}$, translating to high area-based rates of around 10 kg N $ha^{-1}$ $y^{-1}$. When treated with $HCO_3^-$ and/or P, however, *S. squarrosum* showed 40% higher fixation rates compared to *S. palustre*, (F=4.510; $P<0.05$) (Figure 1). Addition of P positively affected $N_2$ fixation for both *Sphagnum* species (F=12.639; $P<0.005$), leading to at least two times higher fixation rates compared to their controls (Figure 1). $HCO_3^-$ addition had an even greater effect, and resulted in around four times higher $N_2$ fixation rates (F=32.103; $P<0.001$) (Figure 1). The combined P and $HCO_3^-$ treatment

increased $N_2$ fixation rate to 300 nmol N $gDW^{-1}$ $h^{-1}$ in *S. squarrosum*.

In general, $N_2$ fixation rates were highest in light incubations and around 10 times lower under dark conditions (F=65.642; $P<0.001$) (Figure 2). However, a similar increase (1.5 times higher) in fixation rates upon P addition was found under both light and dark conditions (F=18.588; $P<0.001$).



### 3.3 Plant performance

*S. squarrosum* and *S. palustre*, had similar photosynthetic rates of around 65 $\mu$mol $CO_2$ $gDW^{-1}$ $h^{-1}$ and showed a strong negative response to $HCO_3^-$-rich water (F=21.468; *P*<0.001), resulting in approximately 50% lower photosynthetic rates (Figure 3). $HCO_3^-$ also resulted in 50-70% lower relative growth rates (F=29.339; *P*<0.001), relative decrease in the number

of capitula (F=86.090; *P*<0.001) and average length (F=268.846; *P*<0.001) of both species (results not shown). Final biomass of $HCO_3$ treated mosses was around 10% lower than that of the control group. Controls of both species ended up with a final dry weight of around 3 g per *Sphagnum* patch, containing around 86 capitula with a length of around 73 mm per moss. This corresponds to a growth rate of 8.5 mg $gDW^{-1}$ $d^{-1}$. In contrast, P treatment did not show an effect on any of the measured plant performance variables of the *Sphagnum* mosses.

### 3.4 Nutrient contents of *Sphagnum*-microorganism association

Concentrations of N, P and K in *Sphagnum* tissue including their microbial community were clearly influenced by surface water treatments (Table 1). Addition of P-rich surface water increased the P content in *Sphagnum*-microorganism tissue by 75% for both *Sphagnum* species (F=11.549; *P*<0.005), while N and K concentrations remained unchanged. In treatments with $HCO_3^-$-rich water the N concentration increased by around 20% (F=6.955; *P*<0.05), and the concentration of K in the

tissue decreased by around 25% (F=140.343; *P*<0.001), without affecting P concentrations, indicating K leakage. Individual N contents did not correlate with $N_2$ fixation rates (results not shown).

N: P ratios differed between the two *Sphagnum* species (F=4.673; *P*<0.05), with overall slightly higher ratios for *S. palustre* (mean of controls: 11.8), compared with *S. squarrosum* (mean controls: 7.9) (Figure 4). These ratios decreased by 57-73%

after addition of P (F=8.656; *P*<0.01) to 6.7 and 5.8 respectively, while $HCO_3^-$ addition did not influence ratios at all. N: K ratios did not differ between the two *Sphagnum* species and were unaffected by addition of P. Addition of $HCO_3^-$ however, increased N: K ratios by 80% (F=143.049; *P*<0.001), due to leaking of K from *Sphagnum* tissue. Therefore the $HCO_3$ treatments were not included in Figure 4.

### 4. Discussion

### 4.1 Diazotrophic activity under high N conditions

Surprisingly, the diazotrophic communities of *S. squarrosum* and *S. palustre* showed appreciable $N_2$ fixation rates of around 40 nmol $N_2$ $gDW^{-1}$ $h^{-1}$, even though they had been subjected to high (25 kg $ha^{-1}$ $y^{-1}$) historical and experimental airborne N input. These rates are well in the range of $N_2$ fixation rates reported by Larmola et al. (2014) for *Sphagnum* spp in Finnish peatlands (0-126 nmol $gDW^{-1}$ $h^{-1}$) and equal to the rates they found for mesotrophic fens, even though atmospheric N inputs

were significantly lower in Finland (3 kg $ha^{-1}$ $y^{-1}$; Mustajärva et al 2008). On an areal basis, $N_2$ fixation rates of our controls




translated to an average N input of 17 kg N ha$^{-1}$ y$^{-1}$ in the upper 2 cm of peat moss for a 250 day growing season (at an average temperature of 13° C). This is in the same order of magnitude as the range of 12-25 kg ha$^{-1}$ y$^{-1}$ reported for pristine boreal bogs, although their growing season only lasts 140 days per year (Vile et al., 2014). Furthermore, similar to Markham (2009), we found *Sphagnum*-associated N$_2$ fixation rates to be at least 5 times higher than those found in feather mosses,

which are around 1.5-3 kg ha$^{-1}$ yr$^{-1}$ (Rousk et al., 2014; DeLuca et al., 2002; Zackrisson et al., 2009; Leppänen et al., 2013). This could be due to morphological differences between the moss species (including hyaline cells of *Sphagnum* providing additional space and protection to microorganisms) and differences in microbial communities resulting from differences in habitat conditions and resources, i.e. availability of inorganic and organic nitrogen and carbon compounds, moisture content and presence of oxygen.

The tissue N concentration of around 11.8 mg g$^{-1}$ in *Sphagnum* spp. appears to be high compared to a range of *Sphagnum* N contents for different N deposition sites (Lamers et al., 2000). Optimal growth conditions for *Sphagnum balticum* were found at an N content of 12.9 mg g$^{-1}$ (Granath et al., 2009), suggesting that *Sphagnum* in our experiment is around the saturation point. Indeed high amounts of inorganic N were still taken up from rainwater by *Sphagnum* spp., leaving the surface water

nearly depleted of N (Table 2). These high N uptake rates, especially for NH$_4^+$, from surface water or rainwater are indeed typical for *Sphagnum* spp. (Fritz et al., 2014). Simultaneously, the associated diazotrophs were still fixing N$_2$ at appreciable rates under these N-rich conditions, even though N$_2$ fixation is an energy demanding process (Vitousek et al., 2002). The fact that N$_2$ fixation rates were high and all N present as NH$_4^+$ in rainwater was taken up by the moss therefore suggests that dissolved inorganic N was not or hardly available for the microbial community and diazotrophs were still experiencing N

limitation. Next to this absolute limitation, the relative lack of N was also great, given the high concentrations of all other (micro)nutrients present in the surface water. So, even the high supply of 25 kg N ha$^{-1}$ y$^{-1}$ by rainwater was rapidly taken up by *Sphagnum*, leaving insufficient N for the microbial community that, in this way, still experienced N limitation.

### 4.2 Role of P availability

*Sphagnum* spp. and their diazotrophic microorganisms were found to respond in a remarkably different way to the addition

of P. As hypothesized, based on N$_2$ fixation being a P demanding process (Vitousek et al., 2002), higher P availability doubled the N$_2$ fixation rates. This increase in N$_2$ fixation by P addition was 75% higher in *Sphagnum squarrosum* compared to S. palustre, pointing out differences in response of the microbiomes of both species. Even more surprising, however, was that *Sphagnum* performance of both species was not at all affected by increased P availability. This implies that diazotrophs were stimulated directly by higher availability of P, rather than indirectly by additional supply of compounds obtained from

the moss. This is also shown by the similar increase of N$_2$ fixation activity with P addition under dark conditions that we found (Figure 2). Most of the diazotrophic activity in both *Sphagnum* species appeared to be light related, as N$_2$ fixation rates went down by 90% under dark conditions. This may have different reasons: 1. most of the diazotrophs are photoautotrophs; 2. most diazotrophs rely on other phototrophic microorganisms for their energy supply; or 3. most diazotrophs depend



directly on products of *Sphagnum* photosynthesis. Since the latter is unlikely given the different response in activity to increased P by *Sphagnum* spp. compared to diazotrophs, the process of $N_2$ fixation, here, seems to depend on phototrophic microorganisms. The high portion of phototrophic microorganisms can be explained by the high availability of nutrients, since mutualistic interactions can be altered by nutrient loading in favor of phototrophic partners (Shantz et al., 2016).

**4.3 Nutrient stoichiometry**

Both in light and dark conditions, diazotrophic activity was increased by P, while *Sphagnum* performance was not. The low N: P ratios of *Sphagnum* tissue of controls (around 10) indicate relative N limitation (Wang and Moore, 2014; Bragazza et al., 2004), which is surprising given the high N loading rates. Although $N_2$ fixation rates doubled, the addition of P resulted in strong accumulation of P in *Sphagnum*-microorganism tissue without additional growth, lowering the N: P ratio to 6 and

pointing towards unbalanced uptake of P or luxury consumption (increased nutrient accumulation without any gain in *Sphagnum* biomass). The amount of N fixed by diazotrophs under light conditions correlates with the N content of *Sphagnum* including its microbiome tissue (Figure 5). When we use the rate of $N_2$ fixation to calculate theoretical increases in N content for different treatments, we find that these indeed explain the increase in N content (result not shown). Still, growth rates remain stable even with increased uptake of P. This unbalanced uptake of P, relative to N, therefore questions a

direct role of the high diazotrophic $N_2$ fixation rates we found here for *Sphagnum* growth, and rather suggests N accumulation in the associated microbial community.

The low N: P ratios seem to be an effect of high P concentrations rather than an indication of N limitation (Jiroušek et al., 2011). As stated before, the absolute N content of *Sphagnum* is high, so N limitation seems unlikely. Concentrations of N, P

and K in *Sphagnum* tissue (including their microbial community) were all high or on the high end for *Sphagnum* in minerotrophic peatlands, particularly for P (Aerts et al., 1999; Lamers et al., 2000; Bragazza et al., 2004) (Table 3). As N: K ratios higher than 3.3 were found to indicate K limitation (Bragazza et al., 2004), the N: K ratios of around 1.6 for the controls in our experiment did not support the idea of K limitation either. Other (micro)nutrients, like Mo were also readily available from the surface water, meaning that most important nutrients did not seem to be limiting *Sphagnum* growth here.

Since light conditions provided in the experiment resulted in at least 80-90% of saturation of the *Sphagnum* photosystem (Harley et al., 1989) and drought was avoided, growth limitation by light or water also seem unlikely. The lack of additional growth with added P and additionally fixed N can therefore most likely be explained by the fact that control peatmosses were already at their physiological maximum. Biomass production, calculated with the average growth rate of 8.5 mg gDW$^{-1}$ d$^{-1}$ and a growth season of 250 days, corresponding to around 300 g m$^{-2}$ y$^{-1}$, was indeed on the high end of production rates

found in literature (250-300 g m$^{-2}$ y$^{-1}$ for various *Sphagnum* species) (Rydin and Jeglum, 2006; Gunnarsson, 2005). The increased $N_2$ fixation rates with the lack of additional biomass production of *Sphagnum* with added P, led to remarkably high amounts of 40 kg ha$^{-1}$ y$^{-1}$ of extra N input in the system.



### 4.4 Both symbiotic partners strongly differ in optimal abiotic conditions

As expected, an increase in $HCO_3^-$ concentration, resulting in a higher alkalinity and related higher pH, decreased *Sphagnum* performance. Photosynthetic rates and relative growth rates decreased by around 50% for both species. Furthermore, $HCO_3^-$ addition led to slightly higher surface water $NH_4^+$ concentrations (Table 2), which most likely resulted from leakage from

*Sphagnum* tissue. Increased N: K ratios indicated that K was also leaking from tissue, both pointing towards cell die-off. This is in accordance with earlier studies that showed sensitivity of *Sphagnum* spp. to buffered conditions (Clymo, 1973; Lamers et al., 1999), although the fen species used in this study are known to be more tolerant than typical bog species (Harpenslager et al., 2015). Here, we showed that direct infiltration of $HCO_3^-$ from mineral-rich surface waters or groundwater into the moss layer negatively affects fen *Sphagnum* spp performance, rather than $Ca^+$, which does not directly

affect pH (Lamers et al., 2015).

To our surprise, the response of the diazotrophic community to high $HCO_3^-$ levels was completely opposite to that of *Sphagnum*. Although *Sphagnum* biomass decreased by 10% after treatment with $HCO_3^-$, the diazotrophic community was stimulated and showed around 4 times higher $N_2$ fixation rates. The increase of $N_2$ fixation may, therefore, have been a direct

effect of leakage of C or other compounds from deteriorating *Sphagnum* tissue. However, a second plausible explanation for the increase in $N_2$ fixation is a direct beneficial effect of the increase in pH in the surface water on microbial growth rates and diazotrophic activity. It is indeed known for aquatic systems that dominant diazotrophs can be inhibited by a decrease in pH (Shi et al., 2012) and from agricultural soils that more diazotrophs are present in higher pH soils (Silva et al., 2013). In addition, the stimulated $N_2$ fixation can be explained by an indirect effect of increased decomposition rates as a result of

buffering (Smolders et al., 2002), leading to the mobilization of additional organic compounds and nutrients from the soil to the surface water. This was also shown in a field gradient analysis at lower atmospheric N-input, where nutrient-rich conditions increased $N_2$ fixation rates (Larmola et al., 2014). Since nutrient concentrations in surface water increased 2 to 5 fold in this study, increased $N_2$ fixation by increased decomposition is a likely third possibility.

Regardless of the effect of $HCO_3^-$ being direct, indirect or both, it is still surprising that diazotrophic microorganisms associated with *Sphagnum*, a genus that requires a low pH and actively acidifies its environment, would thrive under more alkaline conditions. This strongly suggests that for the diazotrophic community the symbiosis with *Sphagnum* seems to be a trade-off, where a sheltered environment (including prevention of drought and predation (Jassey et al., 2013; Andersen et al., 2013)) in hyaline cells outweighs the sub-optimal, acidic conditions and the competition with *Sphagnum* for nutrients.

### 4.5 Importance of the symbiosis

So, in these N rich fen systems, where *Sphagnum* spp. work as a filter monopolizing N and performing well on high nutrient concentrations as long as $HCO_3^-$ concentrations do not become too high, their microbial community still experiences N




limitation. With all N taken up by *Sphagnum*, diazotrophs fix $N_2$ at an appreciable rate despite the high N deposition. These rates are even more increased by addition of P and by a higher $HCO_3^-$ concentration, as an effect of increased pH or an increase of other (micro)nutrients. Diazotrophs seem to have different optimal environmental conditions than their host and seem to trade off shelter from herbivores inside *Sphagnum* hyaline cells against *Sphagnums* monopolization of N and

acidification of the environment. Besides, as peat mosses did not benefit from additional P in combination with the additional N, active control of the diazotrophic community (e.g. by additional organic compound supply) seems unlikely. Given the high $N_2$ fixation rates and accumulation of N in *Sphagnum* peat, we hypothesize that the fixed N is only available by reabsorption from decaying and dead *Sphagnum* tissue and dead microbial biomass, rather than by the direct transfer between diazotrophs and *Sphagnum*. However, this needs to be studied more thoroughly in nutrient limited systems.

Under pristine conditions the high $N_2$ fixation rates of diazotrophs in *Sphagnum* might be important in the N acquisition of these mosses and consequently in the total N cycle of these peatlands, since N accumulation in *Sphagnum* peat is high (Vile et al., 2014). However, while this may well be an essential process to obtain sufficient N for *Sphagnum* growth in pristine systems, we showed that the N fixed seems not to be used for *Sphagnum* growth in high N deposition sites. Rather, the

additionally fixed N is to a large extent stored in *Sphagnum*-microorganism tissue, probably in microbial cells. Most likely only after microorganisms in *Sphagnum* tissue have died off and the N is not further recycled in the microbial community in the hyaline cells, the N is made available for *Sphagnum* by mineralization processes. Different pathways of N transfer between *Sphagnum* and microorganisms were also discussed for $N_2$ fixing methanotrophs by Ho and Bodelier (2015) and also feather mosses were suggested not to depend on their cyanobacteria for N (Rousk and Michelsen, 2016). Since N loads

(25 kg $ha^{-1}$ $y^{-1}$) were high here, and $N_2$ fixation added 10 kg $ha^{-1}$ $y^{-1}$ or more with high P loads, peat mosses might not be able to reabsorb the mineralized N, which then leaches to deeper peat layers. Here, it becomes available to vascular plants that change the functioning of the ecosystem (Lamers et al., 2000). In this way, the high $N_2$ fixation rates might eventually speed up decomposition rates and invasion of vascular plants and thereby the decline of peatlands by supplying additional N to an already N loaded system. Moreover, high input of P still increases $N_2$ fixation rates and therefore, instead of balancing out

the high N loads, they are increased even further.

### 5. Conclusions

1. In N saturated fens with an N deposition of 25 kg $ha^{-1}$ $y^{-1}$ the activity of diazotrophs can still be unexpectedly high (40 nmol N $gDW^{-1}$ $h^{-1}$). Since *Sphagnum* spp. monopolize all N in surface water, its microbial community still experiences N limitation.

2. Diazotrophs are stimulated by addition of P and $HCO_3^-$ (buffer capacity) benefitting from additional organic compounds, nutrients and/or an increase in pH, which explains variations in $N_2$ fixation rates reported for peatlands differing in nutrient supply or buffering.



3. *Sphagnum* growth is -in stark contrast- hampered by the high $HCO_3^-$ concentrations. This questions the concept of a direct mutualism and seems to point to a compromise for the diazotrophic community between a sheltered environment on the one hand and a sub-optimal pH and competition for nutrients with their host on the other.

4. Appreciable $N_2$ fixation rates in *Sphagnum* in high N deposition sites result in a very high total N input, which may speed up decomposition and stimulate the invasion of vascular plants, affecting C sequestration.

**Acknowledgements**

The authors would like to thank Stefan Weideveld for his help with practical work and Paul van der Ven and Jelle Eygensteyn for assisting with the chemical analyses. We also thank Landschap Noord-Holland for approval of collecting soil and plant material from Ilperveld. M.A.R.K. and M.S.M.J. were supported by the ERC (AG EcoMOM; 339880), S.F.H. was supported by an STW grant (PeatCap; 11264), C.F. was supported by a FP7 Grant (Euroot; 289300) and an ERA-NET Plus Action Grant on Climate Smart Agriculture (Cinderella; FP 7 and NWO co-funded) and K.F.E. was supported by a VENI grant (863.13.007) from NWO.

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



**Tabels**

Table 1. Properties of peat monoliths in the experiment (N=16).

|  | Mean | S.E.M. |
|---|---|---|
| Bulk density (kg DW $L^{-1}$) | 0.268 | 0.0101 |
| Organic matter (mg $g^{-1}$) | 573.3 | 28.5 |
| C (mg $g^{-1}$) | 294.7 | 14.5 |
| N (mg $g^{-1}$) | 18.02 | 0.598 |
| P (mg $g^{-1}$) | 0.800 | 0.043 |
| K (mg $g^{-1}$) | 2.00 | 0.156 |




Table 2. Surface water characteristics for the different treatments: control (C), addition of P (P) or $HCO_3^-$ (HCO₃), or both (P + HCO₃). Displayed are means ± standard error of the mean (N=4). Unit for alkalinity (alk) is meq $L^{-1}$, for all elements concentrations are expressed as µmol $L^{-1}$. In the effect row, significant differences of P or HCO₃ treatment are indicated by asterisks, where * represents P≤0.05, ** represents P≤0.01 and *** represents P≤0.001.

|  | pH | alk | NO₃ | NH₄ | P | K | S |
|---|---|---|---|---|---|---|---|
| C | 4.37 | 0.06 | 0.00 | 0.83 | 0.74 | 10.42 | 36.32 |
|  | ± 0.09 | ± 0.03 | ± 0.00 | ± 0.06 | ± 0.36 | ± 1.06 | ± 7.38 |
| P | 4.31 | 0.09 | 0.46 | 0.66 | 5.97 | 9.72 | 30.32 |
|  | ± 0.03 | ± 0.04 | ± 0.27 | ± 0.20 | ± 0.41 | ± 0.30 | ± 8.54 |
| HCO₃ | 7.59 | 2.76 | 0.00 | 3.10 | 3.86 | 11.37 | 102.93 |
|  | ± 0.10 | ± 0.04 | ± 0.00 | ± 0.54 | ± 2.24 | ± 1.10 | ± 57.05 |
| HCO₃ + P | 8.40 | 2.86 | 0.03 | 4.15 | 5.24 | 16.45 | 67.81 |
|  | ± 0.38 | ± 0.08 | ± 0.03 | ± 0.39 | ± 1.38 | ± 2.18 | ± 15.45 |
| P effect |  |  |  |  | * |  |  |
| HCO₃ effect | *** | *** |  | *** |  | * |  |

|  | Al | Ca | Fe | Mg | Mn | Na | Cl |
|---|---|---|---|---|---|---|---|
| C | 6.08 | 25.25 | 7.17 | 16.00 | 0.29 | 113.09 | 26.96 |
|  | ± 1.92 | ± 5.40 | ± 3.53 | ± 2.02 | ± 0.06 | ± 3.31 | ± 2.30 |
| P | 4.86 | 19.28 | 10.94 | 12.54 | 0.23 | 130.32 | 16.02 |
|  | ± 0.50 | ± 6.45 | ± 5.92 | ± 4.36 | ± 0.05 | ± 8.80 | ± 10.73 |
| HCO₃ | 14.65 | 54.99 | 60.32 | 34.16 | 0.54 | 2819.60 | 66.00 |
|  | ± 2.22 | ± 20.32 | ± 6.02 | ± 10.89 | ± 0.18 | ± 72.70 | ± 16.87 |
| HCO₃ + P | 14.92 | 43.03 | 31.18 | 27.52 | 0.39 | 2900.83 | 102.35 |
|  | ± 0.87 | ± 11.33 | ± 9.27 | ± 3.47 | ± 0.03 | ± 94.27 | ± 18.36 |
| P effect |  |  |  |  |  |  |  |
| HCO3 effec | *** | * | *** | ** |  | *** | *** |



Table 3. Concentrations of N, P and K (mg g$^{-1}$) in Sphagnum for different treatments. Since no significant differences between species were found, data of both species were combined to display mean ± standard error (N=8). In effect row, significant differences of P or HCO$_3$ treatment are indicated by asterisks: * P≤0.05, ** P≤0.01 and *** P≤0.001.

| | N (mg g$^{-1}$) | P (mg g$^{-1}$) | K (mg g$^{-1}$) |
|---|---|---|---|
| C | 11.80 | 1.36 | 7.56 |
| | ± 0.53 | ± 0.22 | ± 0.71 |
| P | 12.38 | 2.36 | 9.41 |
| | ± 1.06 | ± 0.38 | ± 1.17 |
| HCO$_3$ | 13.50 | 1.73 | 2.31 |
| | ± 1.19 | ± 0.22 | ± 0.20 |
| HCO$_3$ + P | 16.05 | 2.82 | 2.10 |
| | ± 1.11 | ± 0.31 | ± 0.11 |
| P effect | * | ** | |
| HCO3 effect | ** | | *** |

**Figures**

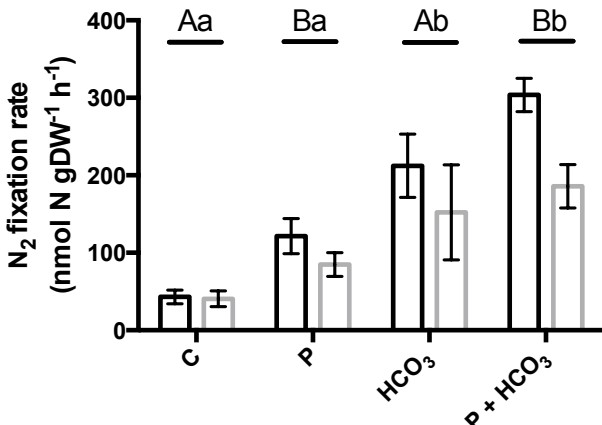

Figure 1. Rates of $N_2$-fixation of the diazotrophic communities of *Sphagnum squarrosum* (dark bars) and *S. palustre* (grey

5    bars), under different treatments. Both P and $HCO_3$ treatment significantly increased $N_2$ fixation in both species, shown by

letter combinations: P treatment (capital letter) and $HCO_3^-$ treatment (lower case).

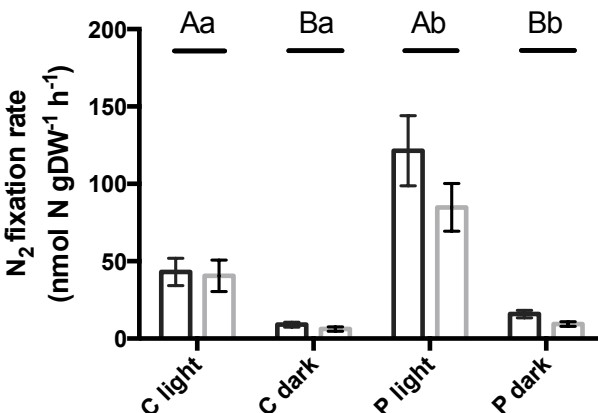

Figure 2. $N_2$ fixation rates of diazotrophic communities of *Sphagnum squarrosum* (dark bars) and *S. palustre* (grey bars),

10    under light or dark conditions. Displayed is the mean ± standard error (N=4) of the control and P treatment (see text). Dark





conditions significantly decreased $N_2$ fixation rates (shown by capital letter) and P treatment significantly increased rates (shown by lower case).

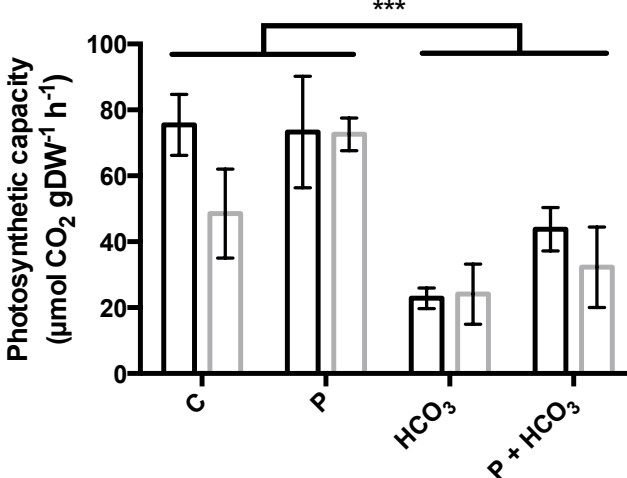

5    Figure 3. Photosynthetic rates of *Sphagnum squarrosum* (dark bars) and *S. palustre* (grey bars) under different surface water treatments. Displayed is the mean ± standard error (N=4). $HCO_3^-$ significantly decreased rates, shown by *** ($P<0.001$).



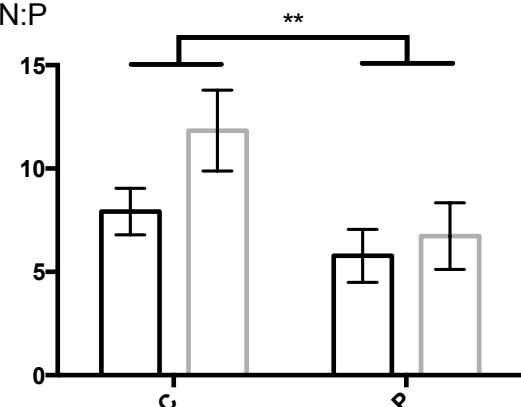

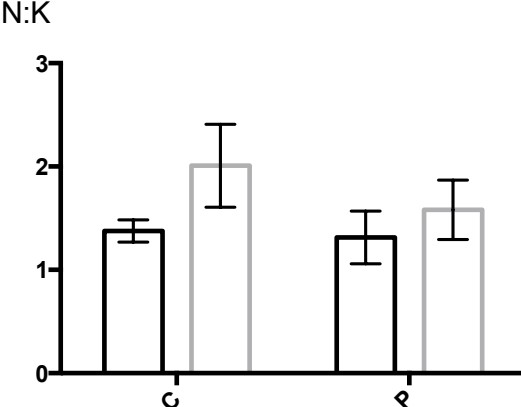

Figure 4. Means of N:P ratio and N:K ratios for *Sphagnum squarrosum* (dark bars) and *S. palustre* (grey bars), displayed for control (C) and addition of P (P) to surface water. Given is the mean ± standard error of the mean (N=4). HCO3 treatments
5  were not included, because of leaking of nutrients from tissue (see text). Significant differences between treatments are shown with ** (P<0.01) in graph.



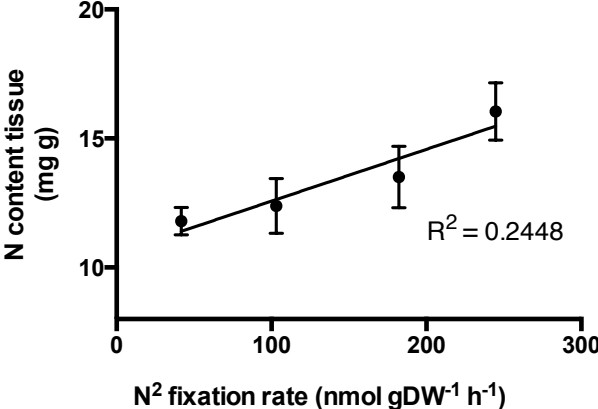

Figure 5. Linear regression of average N content of *Sphagnum* including its microbiome against average $N_2$ fixation rates of both species.