# Peer review of "Symbiosis revisited: phosphorus and acid buffering stimulate $N_2$ fixation but not Sphagnum growth"

_Biogeosciences, 2016_

## Referee Comment (RC1) · Anonymous Referee #1 · 4 Nov 2016

General comments This is an interesting contribution in which the authors demonstrate that nitrogen fixation and plant growth are not affected in the same way by phosphorus or bicarbonate. Their expectation was that increased N fixation, especially when coupled with additional phosphorus, would stimulate sphagnum performance. However, while both phosphorus and bicarbonate increased N fixation, they had neutral or negative effects on sphagnum photosynthesis, respectively. Based on this discrepancy between expected result and the actual result, the authors question the concept of a "direct mutualism".

I am not sure what a "direct mutualism" is. This should be defined.

I wonder whether the effect of P on plant performance is determinable in the time

frame of this study. If, in response to P addition, N fixation increases, if this added N is retained by the microbes, they would have to turn over before the plant could have access to it. Thus, some more time may be necessary to see a P effect on plant performance.

Specific comments The timing of events is not clear. How long were the treatments administered? How much time elapsed between treatment initiation and photosynthesis measurement or nutrient analysis?

The device used to measure photosynthesis is not designed specifically for measuring photosynthesis. One question is whether it is accurate and stable enough to actually detect small but significant differences in photosynthesis between treatments. Also, were the light levels used (not specified) representative of those expected in the field (as opposed to the mesocosm)? If not, the results from this measurement could be irrelevant.

The natural conditions of the site where the mosses were collected should be indicated (pH, bicarbonate concentration, phosphate concentration, etc.) in order to place in context the experimental treatments.

There appears to be a significant interaction between bicarbonate and P with respect to photosynthesis. That is P did not have a significant effect in the absence of bicarbonate, but it did in the presence of bicarbonate. So, when bicarbonate is present, P may be beneficial.

Technical corrections None

---

## Referee Comment (RC2) · Anonymous Referee #2 · 10 Nov 2016

General comments: This paper is of environmental importance as authors have discussed the symbiosis of peat plants and symbiotic microorganisms. They are of recent importance as they play vital role in carbon sequestration. It is an interesting paper as the outcomes obtained were not as obvious expected results. However, there are certain flaws in the approaches they have chosen and discussion made. Moreover, it does not have any broader impacts. Though the methodology is very meticulously designed; some pictures or a graphical abstract would make the approach more clear.

Specific comments: 1. Word "symbiosis" in the title of paper is little ambiguous as the paper is only about the relation of P and N fixation and plant growth. Nowhere the microbial community had been addressed. 2. Abstract is quite general; more specific

results could have been included. 3. Actual field conditions should have been studied and mentioned in the paper. Possibly, few revelations could have been seen like for eg. presence of other growth promoting microorganisms in natural environment which could affect the P/N uptake and plant growth. 4. Time course studies have not been well defined. 5. Three way ANOVA is the statistical technique used here using three independent variable (P, HCO3 and spp.) which is an appropriate technique. But, three way ANOVA is a technique in which dependent variables should be at continuous level. Here, some dependent variables do not come under this assumption. Moreover; the independent variable should have two or more categorical groups. Authors fail to do so. Authors can read:https://statistics.laerd.com/spss-tutorials/three-way-anova-using-spss-statistics.php. Also, post-hoc analysis would make the scenario more clear as it would give precise idea of dependency of each of the independent variable.

Technical comments:Language used in the paper is pretty precise and clear. 1. Number of keywords can be reduced 2. Flow of introduction can be changed. Mention all the required introduction first and then mention your assumptions and reason for doing this study at the end. 3.If your mentioning anything in your paper for first time mention it clearly. Like page 3, line 25, it was mentioned "our field sites"; as it was being mentioned for the first time it is better to mention the name.

---

## Author Comment (AC1) · 1 Dec 2016

We would like to thank the referee for his/her interest and input to the manuscript. We have considered all comments and below you can find our itemized list of responses (b) to the referees comment (a) with changes to the manuscript included (supplement). Pages and lines refer to the revised manuscript with revisions highlighted, which can be found as a supplement file.

General comments:

1a) This is an interesting contribution in which the authors demonstrate that nitrogen fixation and plant growth are not affected in the same way by phosphorus or bicar-

bonate. Their expectation was that increased N fixation, especially when coupled with additional phosphorus, would stimulate sphagnum performance. However, while both phosphorus and bicarbonate increased N fixation, they had neutral or negative effects on sphagnum photosynthesis, respectively. Based on this discrepancy between expected result and the actual result, the authors question the concept of a "direct mutualism". I am not sure what a "direct mutualism" is. This should be defined.

1b) We thank the referee for distinguishing this manuscript as an interesting contribution. With a direct mutualism we refer to a mutualism between microbial symbiont and a plant host, in which the host benefits from a direct transfer of nutrients from the symbiont (bacteria) to the host (Sphagnum), as explained by Ho & Bodelier (2015) (see references manuscript). This is in contrast to the uptake of nutrients after die-off of bacteria, which cannot be regarded as a real and direct mutualism, as this is the case for any plant taking up nutrients from decomposed bacterial biomass.

We agree that this term should be explained to readers and defined more thoroughly in the introduction of the manuscript. Therefore we added the following clarification to Page 3, lines 17-18 and lines 21-23: "...a process that we refer to as a direct mutualism, with reference to the direct transfer of chemicals between host and symbiont (Ho and Bodelier, 2015)" and "There may also be a different, indirect type of interaction in which Sphagnum receives a flow of nutrients from dead and lysed microorganisms."..."i.e. a direct mutualism or an indirect interaction,"

2a) I wonder whether the effect of P on plant performance is determinable in the time frame of this study. If, in response to P addition, N fixation increases, if this added N is retained by the microbes, they would have to turn over before the plant could have access to it. Thus, some more time may be necessary to see a P effect on plant performance.

2b) We thank the referee for bringing up this point, as we can indeed improve our manuscript by elaborating on this.

In various studies on peatlands the stimulating effect of phosphorus on Sphagnum growth was shown. During one growing season, phosphorus was shown to increase photosynthesis by 14% (Fritz et al 2001) and the length of photosynthetic material of Sphagnum significantly by 42% (Carfrae et al., 2007), being 6-7 mm. Although our experiment lasted for 10 weeks, which is shorter than a growing season, it would still be sufficient to be able to see the effects of phosphorus addition on Sphagnum growth, being around 3 mm additional growth.

The additional citations were added on Page 4, line 5: "...and lead to an increase in photosynthesis (by 14%) (Fritz et al., 2012) and moss growth (by 42%) (Carfrae et al., 2007)."

Besides, we expected the mutualistic interaction between Sphagnum and its dia-zotrophs to be of a direct nature (i.e. direct transfer of nutrients between symbiont and host, see previous point) and therefore an increase in NÂň2 fixation rates to result in a direct increase of N availability for Sphagnum, and thus an increase in growth rate. Such a mutualism is for instance known for Azolla spp, were addition of P can in a short time (four weeks or less) greatly increase growth and also N content (Cheng et al., 2010). From our results such a direct symbiotic relationship between Sphagnum and its microbial community seems not to be the case, pointing towards an indirect interaction.

To clarify this point, we added information on Azolla spp to the introduction Page 4, lines 2-3: "...and in Azolla spp, a fern species with symbiotic cyanobacteria within its leaves, P was shown to drastically increase plant growth and N content (Cheng et al., 2010)." And to the discussion we added to Page 11, line 3-6: "In stark contrast with Azolla spp, where P addition is known to directly increase the growth rate and N content of the host plant (Cheng et al., 2010) (direct mutualism), the symbiosis between Sphagnum and its microbial community seems to be based on the indirect transfer of nutrients from microbial die-off (Ho and Bodelier, 2015)."

Specific comments:

3a) The timing of events is not clear. How long were the treatments administered? How much time elapsed between treatment initiation and photosynthesis measurement or nutrient analysis?

3b) We agree that the timing was not stated clearly. The treatments were added during 10 weeks, after which nutrient content, nitrogen fixation and photosynthesis were measured. We added this information to the methods section on Page 5, line 28-29: "Treatment solutions were supplied during ten weeks, after which plant, microbial and abiotic measurements were conducted."

4a) The device used to measure photosynthesis is not designed specifically for measuring photosynthesis. One question is whether it is accurate and stable enough to actually detect small but significant differences in photosynthesis between treatments. Also, were the light levels used (not specified) representative of those expected in the field (as opposed to the mesocosm)? If not, the results from this measurement could be irrelevant.

4b) The device used is a small chamber connected in a closed loop to a near infra red spectroscopy (NIRS) gas analyzer with cavity ring down spectroscopy (CRDS), which is at present the most accurate instrument to measure $CO_2$ fluxes (and $CH_4$, $N_2O$ fluxes), including decreased $CO_2$ levels as a result of photosynthesis. The set-up is therefore exactly similar to, e.g., those using Infrared Spectroscopy Gas Analyzers (IRGA) to measure photosynthesis (Hunt, 2003). The laser-based NIRS-CRDS devices can, however, measure changes in $CO_2$ concentration much faster and with an extremely high resolution (Crosson, 2008). The light levels used were 150 $\mu$mol PAR m-2 s-1, similar to mesocosm and field light levels.

In order to make this more clearly, we added more details on this method from Page 5, line 31 to Page 6, line 4: "...fast greenhouse gas analyzer (NIRS) with cavity ringdown spectroscopy (CRD)" "at similar light conditions as used in the experimental set up (150

$\mu$mol m-2 s-1 PAR)" "...in a closed loop with the NIRS-CRDS gas analyzer capable of measuring concentration changes at a very high resolution (Crosson, 2008) and of accurately measure photosynthesis (Hunt, 2003).

5a) The natural conditions of the site where the mosses were collected should be indicated (pH, bicarbonate concentration, phosphate concentration, etc.) in order to place in context the experimental treatments.

5b) We agree that this information would benefit our manuscript. Therefore we added an additional table, table 1A to Page 20 and referred to it in the method section on Page 5, line 1-2: "Field conditions of the site where the mosses were collected are shown in Table 1A."

6a) There appears to be a significant interaction between bicarbonate and P with respect to photosynthesis. That is P did not have a significant effect in the absence of bicarbonate, but it did in the presence of bicarbonate. So, when bicarbonate is present, P may be beneficial.

6b) We are very grateful that the referee noticed this error. Apparently, the wrong figure was added to the manuscript, which we very much regret and apologize for. We changed this figure for the correct one, in which there is no appearance of an interaction effect, in agreement with the statistics, as stated on Page 7, line 23-24: "No interaction effects were found for any of the parameters".

Technical corrections: None

Additional references:

Carfrae, J., Sheppard, L., Raven, J., Leith, I., and Crossley, A.: Potassium and phosphorus additions modify the response of Sphagnum capillifolium growing on a Scottish ombrotrophic bog to enhanced nitrogen deposition, Applied Geochemistry, 22, 1111-1121, 2007.

Cheng, W. G., Sakai, H., Matsushima, M., Yagi, K., and Hasegawa, T.: Response of the

floating aquatic fern Azolla filiculoides to elevated CO2, temperature, and phosphorus levels, Hydrobiologia, 656, 5-14, 2010.

Crosson, E. R.: A cavity ring-down analyzer for measuring atmospheric levels of methane, carbon dioxide, and water vapor, Applied Physics B, 92, 403-408, 2008.

Fritz, C., Van Dijk, G., Smolders, A. J. P., Pancotto, V. A., Elzenga, T. J. T. M., Roelofs, J. G. M., and Grootjans, A. P.: Nutrient additions in pristine Patagonian Sphagnum bog vegetation: can phosphorus addition alleviate (the effects of) increased nitrogen loads, Plant Biology, 14, 491-499, 2012.

Ho, A., and Bodelier, P. L. E.: Diazotrophic methanotrophs in peatlands: the missing link?, Plant and Soil, 389, 419-423, 2015.

Hunt, S.: Measurements of photosynthesis and respiration in plants, Physiologia Plantarum, 117, 314-325, 2003.

Please also note the supplement to this comment:
http://www.biogeosciences-discuss.net/bg-2016-384/bg-2016-384-AC1-supplement.pdf

**Supplement:**

[revised manuscript text omitted]

van den Elzen 12/1/16 1:16 PM

---

## Author Response (AR1)

Dear editor,

We would like to thank you for your comments on our revised manuscript.

Based on your recommendation, we further revised and improved the discussion. In addition to the changes that we had already made in response to the referees' comments, we made the following improvements to the discussion:

- In order to improve clarity, we changed the order of sections to a more logical sequence, discussing the effects of bicarbonate first;
- We merged the sections on phosphorus addition and nutrient stoichiometry;
- The section on the role of phosphorus availability may have been confusing, and the change in the order of paragraphs now leads to a more logical build up of our arguments;
- The last section on the importance of the symbiosis was shortened to improve the focus on our results.

Below, we added an itemized list of all changes we made since the publishing of the manuscript as a discussion paper. First you will find our responses to the comments of the referees as they were published online earlier, with adapted references to the lines of the new version of the manuscript. After this, we provided a list of the additional changes we made based on your decision for major revision.

Yours sincerely, on behalf of all authors,

Eva van den Elzen

**General comments referee 1:**

1a) This is an interesting contribution in which the authors demonstrate that nitrogen fixation and plant growth are not affected in the same way by phosphorus or bicarbonate. Their expectation was that increased N fixation, especially when coupled with additional phosphorus, would stimulate sphagnum performance. However, while both phosphorus and bicarbonate increased N fixation, they had neutral or negative effects on sphagnum photosynthesis, respectively. Based on this discrepancy between expected result and the actual result, the authors question the concept of a "direct mutualism". I am not sure what a "direct mutualism" is. This should be defined.

*1b) We thank the referee for distinguishing this manuscript as an interesting contribution. With a direct mutualism we refer to a mutualism between microbial symbiont and a plant host, in which the host benefits from a direct transfer of nutrients from the symbiont (bacteria) to the host (Sphagnum), as explained by Ho & Bodelier (2015) (see references manuscript). This is in contrast to the uptake of nutrients after die-off of bacteria, which cannot be regarded as a real and direct mutualism, as this is the case for any plant taking up nutrients from decomposed bacterial biomass.*

*We agree that this term should be explained to readers and defined more thoroughly in the introduction of the manuscript. Therefore we added the following clarification to Page 3, lines 19-20 and lines 23-25: "...a process that we refer to as a direct mutualism, with reference to the direct transfer of chemicals between host and symbiont (Ho and Bodelier, 2015)" and "There may also be a different, indirect type of interaction in which Sphagnum receives a flow of nutrients from dead and lysed microorganisms."..."i.e. a direct mutualism or an indirect interaction,"*

2a) I wonder whether the effect of P on plant performance is determinable in the time frame of this study. If, in response to P addition, N fixation increases, if this added N is retained by the microbes, they would have to turn over before the plant could have access to it. Thus, some more time may be necessary to see a P effect on plant performance.

*2b) We thank the referee for bringing up this point, as we can indeed improve our manuscript by elaborating on this.*

*In various studies on peatlands the stimulating effect of phosphorus on Sphagnum growth was shown. During one growing season, phosphorus was shown to increase photosynthesis by 14% (Fritz et al 2001) and the length of photosynthetic material of* Sphagnum *significantly by 42% (Carfrae et al., 2007), being 6-7 mm. Although our experiment lasted for 10 weeks, which is shorter than a growing season, it would still be sufficient to be able to see the effects of phosphorus addition on* Sphagnum *growth, being around 3 mm additional growth.*

*The additional citations were added on Page 4, line 7: "...and lead to an increase in photosynthesis (by 14%) (Fritz et al., 2012) and moss growth (by 42%) (Carfrae et al., 2007)."*

*Besides, we expected the mutualistic interaction between Sphagnum and its diazotrophs to be of a direct nature (i.e. direct transfer of nutrients between symbiont and host, see previous point) and therefore an increase in $N_2$ fixation rates to result in a direct increase of N availability for Sphagnum, and thus an increase in growth rate. Such a mutualism is for instance known for* Azolla *spp, were addition of P can in a short time (four weeks or less) greatly increase growth and also N content (Cheng et al., 2010). From our results such a direct symbiotic relationship between* Sphagnum *and its microbial community seems not to be the case, pointing towards an indirect interaction.*

*To clarify this point, we added information on* Azolla *spp to the introduction Page 4, lines 4-5: "...and in* Azolla *spp, a fern species with symbiotic cyanobacteria within its leaves, P was shown to drastically increase plant growth and N content (Cheng et al., 2010)."*
*And to the discussion we added to Page 12, line 14-17: "This is in stark contrast to* Azolla *spp, where P addition is known to directly increase the growth rate and N content of the host plant (direct mutualism) (Cheng et al., 2010). Under the present environmental conditions, the symbiosis between Sphagnum and its microbial community seems to be based on the indirect transfer of nutrients from microbial die-off (Ho and Bodelier, 2015)."*

**Specific comments referee 1:**
3a) The timing of events is not clear. How long were the treatments administered? How much time elapsed between treatment initiation and photosynthesis measurement or nutrient analysis?

*3b) We agree that the timing was not stated clearly. The treatments were added during 10 weeks, after which nutrient content, nitrogen fixation and photosynthesis were measured. We added this information to the methods section on Page 6, line 1-2: "Treatment solutions were supplied during ten weeks, after which plant, microbial and abiotic measurements were conducted."*

4a) The device used to measure photosynthesis is not designed specifically for measuring photosynthesis. One question is whether it is accurate and stable enough to actually detect small but significant differences in photosynthesis between treatments. Also, were the light levels used (not specified) representative of those expected in the field (as opposed to the mesocosm)? If not, the results from this measurement could be irrelevant.

*4b) The device used is a small chamber connected in a closed loop to a near infra red spectroscopy (NIRS) gas analyzer with cavity ring down spectroscopy (CRDS), which is at present the most accurate instrument to measure $CO_2$ fluxes (and $CH_4$, $N_2O$ fluxes), including decreased $CO_2$ levels as a result of photosynthesis. The set-up*

*is therefore exactly similar to, e.g., those using Infrared Spectroscopy Gas Analyzers (IRGA) to measure photosynthesis (Hunt, 2003). The laser-based NIRS-CRDS devices can, however, measure changes in $CO_2$ concentration much faster and with an extremely high resolution (Crosson, 2008). The light levels used were 150 μmol PAR $m^{-2}$ $s^{-1}$, similar to mesocosm and field light levels.*

*In order to make this more clearly, we added more details on this method from Page 6, line 4 to line 9: "...fast greenhouse gas analyzer (NIRS) with cavity ringdown spectroscopy (CRD)" "at similar light conditions as used in the experimental set up (150 μmol $m^{-2}$ $s^{-1}$ PAR)" "...in a closed loop with the NIRS-CRDS gas analyzer capable of measuring concentration changes at a very high resolution (Crosson, 2008) and of accurately measuring photosynthesis (Hunt, 2003).*

5a) The natural conditions of the site where the mosses were collected should be indicated (pH, bicarbonate concentration, phosphate concentration, etc.) in order to place in context the experimental treatments.

*5b) We agree that this information would benefit our manuscript. Therefore we added an additional table, table 1A to Page 18 and referred to it in the method section on Page 5, line 5-6: "Field conditions of the site where the mosses were collected are shown in Table 1A."*

6a) There appears to be a significant interaction between bicarbonate and P with respect to photosynthesis. That is P did not have a significant effect in the absence of bicarbonate, but it did in the presence of bicarbonate. So, when bicarbonate is present, P may be beneficial.

*6b) We are very grateful that the referee noticed this error. Apparently, the wrong figure was added to the manuscript, which we very much regret and apologize for. We changed this figure for the correct one on Page 23, in which there is no appearance of an interaction effect, in agreement with the statistics, as stated on Page 7, line 27-28: "No interaction effects were found for any of the parameters".*

**General comments referee 2**:

1a) This paper is of environmental importance as authors have discussed the symbiosis of peat plants and symbiotic microorganisms. They are of recent importance as they play vital role in carbon sequestration. It is an interesting paper as the outcomes obtained were not as obvious expected results. However, there are certain flaws in the approaches they have chosen and discussion made. Moreover, it does not have any broader impacts. Though the methodology is very meticulously designed; some pictures or a graphical abstract would make the approach more clear.

*1b) We thank the referee for the interest and input with respect to the manuscript, and the statement that our paper is of environmental importance. We do indeed believe that our results have broader impacts, i.e. that the regulation of nitrogen*

*fixation by phosphorus is essential for our understanding of the nitrogen cycle, and how it influences the sequestration of carbon in peatlands. Besides, high additional doses of nitrogen by phosphorus-induced nitrogen fixation to already nitrogen-loaded peatlands can well be expected to lead to serious degradation of these important C storing systems. This is important in the context of ecosystem restoration in high nitrogen areas in which the input of phosphorus is simultaneously abundant and not able to offset nitrogen loads as could be expected.*

*We also thank the reviewer for the idea of adding a graphic figure of the experimental set up.  We added both a picture and a graphical representation to new Figure 1 on Page 21. We referred to this picture in the methods section on Page 5, lines 26-27: "A graphic figure of the experimental set up and photo's can be found in Figure 1."*

**Specific comments referee 2:**

2.1a) Word "symbiosis" in the title of paper is little ambiguous as the paper is only about the relation of P and N fixation and plant growth. Nowhere the microbial community had been addressed.

*2.1b) We did indeed not assess the full microbial community, because that was not the purpose of our research. However, we did study the activity of nitrogen fixing microorganisms, and how this affects* Sphagnum *growth. We choose to keep the word symbiosis in the title of the manuscript, because the interaction between the host* Sphagnum *and its diazotrophic microbiome is central to this manuscript.*

2.2a) Abstract is quite general; more specific results could have been included.

*2.2b) In attempt to keep the abstract sufficiently concise, we decided to only include our main results. To be more specific, we have now added the exact rates of nitrogen fixation and the results of the light compared to dark incubations for nitrogen fixation to Page 2, line 7: "at a rate of 40 nmol N gDW$^{-1}$ h$^{-1}$" and to Page 2, line 14-15: "In addition, nitrogen fixation was found to strongly depend on light, with rates 10 times higher in light conditions suggesting high reliance on phototrophic organisms for carbon."*

2.3a) Actual field conditions should have been studied and mentioned in the paper. Possibly, few revelations could have been seen like for eg. presence of other growth promoting microorganisms in natural environment which could affect the P/N uptake and plant growth.

*2.3b) We thank the reviewer for this remark and agree that the context of the field conditions would benefit our manuscript. Therefore, we added a table (Table 1A) with the abiotic conditions of the field site where the mosses were collected to Page*

*18 and referred to it in the method section on Page 5, line 5-6: "Field conditions of the site where the mosses were collected are shown in Table 1A."*

*However, additional information on the microbial community of peat soil we did not assess, since this would be out of our scope, very elaborate, and a different study by itself.*

2.4a) Time course studies have not been well defined.

*2.4b) We thank the reviewer for noticing this. The time course of the experiment was 10 weeks. We added this information to the methods section on Page 6, line 1-2: "Treatment solutions were supplied during ten weeks, after which plant, microbial and abiotic measurements were conducted."*

2.5a) Three way ANOVA is the statistical technique used here using three independent variable (P, HCO3 and spp.) which is an appropriate technique. But, three way ANOVA is a technique in which dependent variables should be at continuous level. Here, some dependent variables do not come under this assumption. Moreover; the independent variable should have two or more categorical groups. Authors fail to do so. Authors can read: f Also, post-hoc analysis would make the scenario more clear as it would give precise idea of dependency of each of the independent variable.

*2.5b) All dependent variables assessed with three-way ANOVA are at a continuous level, including nitrogen fixation rate, relative growth rate, number of capitula, length increment, pore water nutrients, alkalinity. Since there are only two groups for each variable, post-hoc analyses cannot be applied. All independent variables have two categorical groups: i.e. +/- phosphorus, +/- bicarbonate,* S. palustre *or* S. squarrosum.

*For clarification, we added this information to the method section on Page 7, line 22-25: "...independent variables (fixed factors) with two categorical groups. All dependent variables were quantitative and at a continuous scale, i.e. nitrogen fixation rate, photosynthetic activity, relative growth rate, number of capitula, Sphagnum length increment, and pore water and tissue nutrient concentrations."*

**Technical comments referee 2:**

3a) Language used in the paper is pretty precise and clear.

*3b) We thank the reviewer for this comment.*

3.1a) Number of keywords can be reduced

*3.1b) We removed 'ecophysiology' and changed 'nutrients' and 'nitrogen deposition' to 'nitrogen' on Page 2, line 26.*

3.2a) Flow of introduction can be changed. Mention all the required introduction first and then mention your assumptions and reason for doing this study at the end.

*3.2b) We have now made changes in order to move all hypotheses to the last paragraph of the introduction, as suggested by the reviewer.*

*On Page 4, lines 7-9 we adapted: "It is therefore expected that the addition of P can improve...N deposition areas". The next sentences of this paragraph "In addition...becomes limiting" we moved to Page 4, lines 25-28. In this last paragraph, we also made adjustments to lines 24-25 and 28-29. Leading to a changed last paragraph in lines 23-30: "Our prime research question was whether P availability and alkalinity were key regulators of both diazotrophic and* Sphagnum *activity, with P increase having a positive effect on both partners, and alkalinity increase a negative effect. In addition, in view of a direct mutualistic relationship between the moss and its diazotrophs, as with* Azolla *spp and its cyanobacteria, we expect that higher $N_2$ fixation rates provide additional N. Combined with higher P availability, this may increase* Sphagnum *photosynthesis and growth even further, as long as no other resource or condition becomes limiting. By testing this hypotheses, we are able to explore the nature of the symbiotic interaction, i.e. which benefits or costs the diazotrophic microbial community experience through the close association with their host, and vice versa."*

3.3a) If your mentioning anything in your paper for first time mention it clearly. Like page 3, line 25, it was mentioned "our field sites"; as it was being mentioned for the first time it is better to mention the name.

*3.3b) We added the specifics of our field site to this line, now on Page 3, line 31.*

**Additional changes based on referee reports**

*1) In order to make the abstract more specific, we changed on Page 2, line 22-24: "concept of a direct mutualism" to "...regulation of nitrogen fixation by* Sphagnum *under these eutrophic conditions. The high $N_2$ fixation rates result in high additional nitrogen loading of 10 kg ha$^{-1}$ y$^{-1}$ on top of the high nitrogen deposition in these ecosystems."*

*2) We further improved the introduction by adding to Page 4, line 28-29: "...we are able to not only investigate regulation of nitrogen fixation by these abiotic factors but also..."*

*3) To improve the discussion, we moved the section "4.3 Both symbiotic partners...*Sphagnum *for nutrients" to Page 10, line 13 – Page 11, line 8 and changed its number to: "4.2". Besides, we made minor changes to Page 10, on line 30: from "more diazotrophs are present" to "diazotrophic communities are larger" and on line 32 from "$N_2$ fixation can be explained" to "$N_2$ fixation may be*

*explained".   On Page 11, line 1, we adapted: "…nutrient-rich conditions correlated with increased $N_2$ fixation rates…"*

*4) The section 'Role of P availability' was numbered '4.3' on Page 11, line 9. From its first paragraph we removed from Page 11, line 23 "Since the latter is unlikely given the different response in activity to increased P by* Sphagnum *spp. compared to diazotrophs, the process of $N_2$ fixation, here, seems to depend on phototrophic microorganisms." And changed the last sentence to: "A high abundance of phototrophic organisms could be…"*

*5) From Page 11, line 24 the title section "4.3 Nutrient stoichiometry" was removed and the first sentence "Both in light…performance was not." was replaced with: "P addition did, however, not increase* Sphagnum *growth, raising the question which other factor may have been limiting its growth." on Page 11, Line 24. Besides the last part of the next sentence was removed: ", which is surprising given the high N loading rates."*

*6) To improve the flow of the discussion, the last paragraph of section 4.3 "The low N:P ratios…input in the system" was inserted in the second paragraph on Page 11, line 27, after "P addition did…(Bragazza et al., 2004)". The first sentence of the inserted part was changed to: "However, under these eutrophic conditions with high N availability and high tissue N concentrations, low ratios rather seem to be an effect of high P concentrations (Jirousek et al., 2011)." And this new paragraph was divided after "…(Rydin and Jeglum, 2006; Gunnarsson, 2005)" on Page 12, line 3. The next sentence: "The increased $N_2$ fixation rates…N input in the system." was removed.*

*7) To shorten the discussion we removed from Page 11, line 27: "As stated before, the absolute N content of* Sphagnum *is high, so N limitation seems unlikely." and from Page 11, line 29-30 we removed: "As N: K ratios higher than 3.3 were found to indicate K limitation". This was changed to: "N: K ratios of around 1.6 for the controls in our experiment did not support the idea of K limitation (Bragazza et al., 2004)." On Page 11, line 31 ",meaning that most important nutrients did not seem to be limiting* Sphagnum *growth here." was removed. Besides, for clarity "Mo" on page 11, line 31 was changed to "molybdenum" and the last sentence of the paragraph on Page 12, line 2-3 was changed to "Biomass production rates (based on the average growth rate…of 250 days) corresponded to around 300 g m$^{-2}$ y$^{-1}$, which is indeed high (Rydin and Jeglum, 2006…)."*

*8) To put the message more clearly the first sentences of the new last paragraph of section 4.3 on Page 12, line 5-6 were changed from "Although $N_2$ fixation rates doubled, the addition of P resulted in strong accumulation of P…" to: "With apparently no nutrient limitation for* Sphagnum *growth, P addition led to accumulation in* Sphagnum*-microorganism tissue. This lowered the N: P ratio, pointing towards…". We also made small changes to Page 12, line 9 to "…different treatments, these can explain…" and removed from Page 12, line 15: "Still, growth rates remain stable even with increased uptake of P. This unbalanced uptake…", instead leaving only the second sentence, changed to: "The unbalanced uptake…"*

*9) To conclude this new section we added after "associated microbial community."
on Page 12, line 11: "In conclusion, either the fixed N was not directly available for*
Sphagnum*, or it could not be used due to physiological constraints. In both cases,*
Sphagnum *could not profit from the additionally fixed N and seemed to be
competing for nutrients with its symbionts rather than regulating their activity by
supplying additional C." To the last sentence of this paragraph on Page 12, line 17,
we added: "...rather than by a mutualistic interaction with* Sphagnum *directly
benefitting from the additionally fixed N. More research is, however, needed to
determine whether the symbiosis would change to a mutualistic interaction at low
N conditions. At the ecosystem level, the increased $N_2$ fixation rates with the lack of
additional biomass production of* Sphagnum *with added P, led to remarkably high
amounts of 40 kg $ha^{-1}$ $y^{-1}$ of extra N input."*

*10) Section 'Importance of the symbiosis' was numbered 4.4 on Page 12, line 22. To
improve this section we added to Page 12, line 26: "or an increase of
(micro)nutrients, other than P. This may well explain the differences in $N_2$ fixation
rates between fens and bogs (Larmola et al., 2014)." Moreover, the two paragraphs
of section 4.4 were merged and "However, this needs to be studied....by
mineralization processes" was removed from Page 12, line 33. Besides, minor
textual changes were made throughout the paragraph from Page 12, line 22 to

[revised manuscript text omitted]

van den Elzen 12/1/16 1:16 PM